# Characterization of a CXCR4 antagonist TIQ-15 with dual tropic HIV entry inhibition properties

**Zheng Zhou**[1], **Jia Guo**[1], **Brian Hetrick**[1], **Sameer Tiwari**[1], **Amrita Haikerwal**[1], **Yang Han**[1], **Vincent C. Bond**[2], **Ming B. Huang**[2], **Marie K. Mankowski**[3], **Beth A. Snyder**[3], **Priscilla A. Hogan**[3], **Savita K. Sharma**[4], **Dennis C. Liotta**[4], **Terry-Elinor Reid**[1], **Lawrence J. Wilson**[4]*, **Yuntao Wu**[1]*

1 Center for Infectious Disease Research, George Mason University, Manassas, Virginia, United States of America, 2 Department of Microbiology, Biochemistry and Immunology, Morehouse School of Medicine, Atlanta, Georgia, United States of America, 3 Department of Infectious Disease Research, Southern Research Institute, Frederick, Maryland, United States of America, 4 Department of Chemistry, Emory University, Atlanta, Georgia, United States of America

* ljwilso@emory.edu (LJW); ywu8@gmu.edu (YW)

**Data Availability Statement:** All data are available in the article and supporting information.

**Funding:** This project was conducted in part by Southern Research Institute using federal funds

## Abstract

The chemokine co-receptors CXCR4 and CCR5 mediate HIV entry and signal transduction necessary for viral infection. However, to date only the CCR5 antagonist maraviroc is approved for treating HIV-1 infection. Given that approximately 50% of late-stage HIV patients also develop CXCR4-tropic virus, clinical anti-HIV CXCR4 antagonists are needed. Here, we describe a novel allosteric CXCR4 antagonist TIQ-15 which inhibits CXCR4-tropic HIV-1 infection of primary and transformed CD4 T cells. TIQ-15 blocks HIV entry with an $IC_{50}$ of 13 nM. TIQ-15 also inhibits SDF-1α/CXCR4-mediated cAMP production, cofilin activation, and chemotactic signaling. In addition, TIQ-15 induces CXCR4 receptor internalization without affecting the levels of the CD4 receptor, suggesting that TIQ-15 may act through a novel allosteric site on CXCR4 for blocking HIV entry. Furthermore, TIQ-15 did not inhibit VSV-G pseudotyped HIV-1 infection, demonstrating its specificity in blocking CXCR4-tropic virus entry, but not CXCR4-independent endocytosis or post-entry steps. When tested against a panel of clinical isolates, TIQ-15 showed potent inhibition against CXCR4-tropic and dual-tropic viruses, and moderate inhibition against CCR5-tropic isolates. This observation was followed by a co-dosing study with maraviroc, and TIQ-15 demonstrated synergistic activity. In summary, here we describe a novel HIV-1 entry inhibitor, TIQ-15, which potently inhibits CXCR4-tropic viruses while possessing low-level synergistic activities against CCR5-tropic viruses. TIQ-15 could potentially be co-dosed with the CCR5 inhibitor maraviroc to block viruses of mixed tropisms.

## Author summary

HIV uses the chemokine co-receptors CXCR4 or CCR5 for cell fusion and entry. While the CCR5-tropic viruses predominate early in HIV infection, the emergence of the

from the Division of AIDS, NIAID/NIH under contract HHSN272201400010I entitled "In Vitro Testing Resources for HIV Therapeutics and Topical Microbicides. This work was also funded in part by George Mason University internal research fund (to Y. W.), and by National Institute of Allergy and Infectious Diseases Grant 1R01AI148012 (to Y.W.). The funders had no role in study design, data collection and analysis, decision to publish, or preparation of the manuscript.

**Competing interests:** The authors have declared that no competing interests exist.

CXCR4-tropic viruses at later stages in 50% of patients is associated with rapid disease progression. The CCR5 antagonist maraviroc has been approved for treating HIV infection. However, currently, there is no clinical anti-HIV CXCR4 inhibitor. Here we report a novel CXCR4 antagonist TIQ-15 that potently blocked CXCR4-tropic HIV infection of human CD4 T cells. TIQ-15 in combination with maraviroc also demonstrated synergistic activities against CCR5-tropic viruses and could be potentially used with maraviroc to block viruses of mixed tropisms.

## Introduction

HIV infection is initiated by simultaneous binding of the viral envelope glycoprotein gp120 to CD4 [1–3] and one of the chemokine co-receptors, CXCR4 [4] or CCR5 [5–10]. The interaction between gp120 V3 loop and the co-receptors mediates viral fusion and entry, and also determines whether the virus uses CCR5, CXCR4 or both, which is frequently referred to as M-tropism, T-tropism or dual-tropism, respectively [11]. The alternative common nomenclature of R5-tropism, X4-tropism or X4R5-tropism is also used. In early infection, M-tropic virus predominates, and mainly uses CCR5 for entry and infection of CCR5+ memory CD4 + T cells [12–14]. In late stages, in 50% of patients, the virus shifts tropism towards the syncytium-inducing T-tropic virus using CXCR4 [15–17]. This tropism switch is associated with rapid disease progression; individuals who are directly infected with T-tropic virus also progress to AIDS more rapidly than M-tropic-infected individuals, supporting an important role of CXCR4 in disease progression [16,18–20].

In addition to mediating viral entry, HIV gp120-CXCR4 interaction also triggers the activation of multiple signaling molecules such as cofilin, LIMK, WAVE2, Pyk2, and Akt (for a review, see [21]). Previous studies have demonstrated that HIV-mediated signaling through CXCR4 is critical for latent infection of blood resting CD4+ T cells [22]. For instance, gp120 binding to CXCR4 on blood resting T cells activates the Rac1-PAK-LIMK-cofilin pathway [22,23] and the WAVE2-Arp2/3 pathway [24], promoting actin dynamics necessary for HIV entry and intracellular migration. To some extent, HIV-triggered signaling through CXCR4 resembles the chemotactic responses mediated by SDF-1 (stromal cell-derived factor 1), the natural ligand of CXCR4, suggesting that HIV-mediated signal transduction in T cells may alter cellular chemotactic responses and cause pathogenesis [25–27]. Indeed, in HIV infected patients, cofilin is found to be hyperactivated in the blood CD4+ T cells [25,26,28]. This HIV-mediated cofilin hyperactivation is correlated with the degree of T cell damages, and inversely correlated with T cell recovery from ART [25].

Given the critical role of chemokine co-receptors in HIV entry, small molecule inhibitors of CCR5 and CXCR4 have been in various stages of development for treating HIV infections [29,30]. Currently, maraviroc is the only chemokine and CCR5 inhibitor that has been approved by both the FDA and the European Medicines Agency (EMA) for HIV treatment [31,32]. However, two other CCR5 inhibitors are progressing through clinical trials (PRO140 and TBR652) [33]. Despite their effectiveness, maraviroc and all future CCR5-based drugs require tropism pre-testing (TROFILE; Monogram Biosciences) to confirm CCR5-tropism before prescription [34], as CCR5 inhibitors may push pre-existing viral tropisms towards the more pathogenic CXCR4-tropic HIV species [35]. Several CXCR4 inhibitors, such as AMD3100 (Plerixafor) and AMD11070, have also been developed and extensively studied. Plerixafor was investigated initially as an anti-HIV drug, but ultimately was approved as an agent for stem cell mobilization [36]. Clinical studies of AMD11070 also showed efficacy in

reducing viral load of T-tropic viruses, however, further studies were suspended due to hepato-toxicity [37]. Currently, there are no other anti-HIV agents targeting CXCR4 that are being investigated in the clinic. However, combination therapy utilizing a CXCR4 inhibitor and a CCR5 inhibitor such as maraviroc would offer a broader inhibition of viruses with all three tropisms (R5, X4, and X4R5).

Herein, we describe the characterization of a novel CXCR4 antagonist TIQ-15, which belongs to a unique subclass of tetrahydroisoquinolines (TIQs) [38]. TIQ-15 exhibited favor-able ADME properties in preliminary tests, lending itself as a promising drug candidate. In particular, TIQ-15 contains a 1,2,3,4-tetrahydroisoquinoline ring, which was found to have minimal inhibitory effect against CYP450 enzymes, unlike its analog AMD11070. We selected TIQ-15 for further mechanistic studies largely due to its high anti-HIV potency and favorable metabolic profile. In this report, we demonstrated that TIQ-15 blocks SDF-1 binding, signal transduction, cofilin activation, and SDF-1-induced T cell chemotaxis. TIQ-15 also downmo-dulated surface CXCR4 density in a dosage-dependent manner (10 nM to 10 μM). TIQ-15 inhibited X4-tropic HIV infection with an $IC_{50}$ of 13 nM without detectable cytotoxicity. Fur-thermore, we demonstrate that TIQ-15 exhibited favorable potency against a panel of clinical HIV-1 isolates with X4-, R5-, and dual-tropism, and acted synergistically with maraviroc in inhibiting R5-tropic HIV entry.

## Results

### TIQ-15 is a Gαi-based CXCR4 antagonist

TIQ-15 was identified as the most potent inhibitor in our initial screening of the TIQ series compounds for blocking HIV on our Rev-dependent GFP indicator CD4 T cell [39,40] (**Figs 1A and S1**). In our previous studies on the TIQ series, we demonstrated that TIQ-15 displays antagonist activity in a calcium flux assay and displaces $^{125}$I-SDF-1 in a dose-dependent fash-ion [38,41]. The question remained whether TIQ-15 displayed characteristics consistent with CXCR4 receptor antagonism, specifically through the Gαi pathway, which is considered the main component of HIV-1 V3 and SDF-1α-based signaling. CXCR4 is a Gαi GPCR, hence, agonism of the receptor would cause a reduction of cAMP levels as a result of inhibition of adenylate cyclase. To measure possible effects on cAMP levels, we utilized the Glow sensor cAMP assay that was previously reported to determine Gαi signaling inhibition after forskolin stimulation of intracellular levels of cAMP [42]. SDF-1α-induced signaling reduced the forsko-lin pre-stimulated cAMP levels in a dose-dependent manner with an $EC_{50}$ of 7.1 nM (**Fig 1B**). However, TIQ-15 blocked SDF-1α-induced cAMP reduction in a dose-dependent fashion with an $IC_{50}$ value of 41 nM compared to that of AMD3100 (347 nM) (**Fig 1B**). These assays confirm that TIQ-15 effectively blocks CXCR4 receptor function as an antagonist, and is more potent than AMD3100.

### TIQ-15 interferes with SDF-1-mediated chemotaxis and cofilin activation in blood resting CD4 T cells

It has been shown that blocking CXCR4 Gαi signaling through pertussis toxin (PTX) inhibits the activation of cofilin, an actin depolymerization factor regulating T cell chemotaxis [22]. Thus, blocking Gαi signal with a CXCR4 antagonist should also prevent cofilin activation and inhibit CXCR4-mediated chemotaxis. It is likely that SDF-1α/CXCR4-mediated chemotactic actin dynamics are directly regulated by cAMP, which can affect downstream phosphorylation events through kinases such as the cofilin kinase LIMK (LIM domain kinase) [23]. Indirectly, cAMP can also regulate secondary messenger proteins such as protein phosphatases via

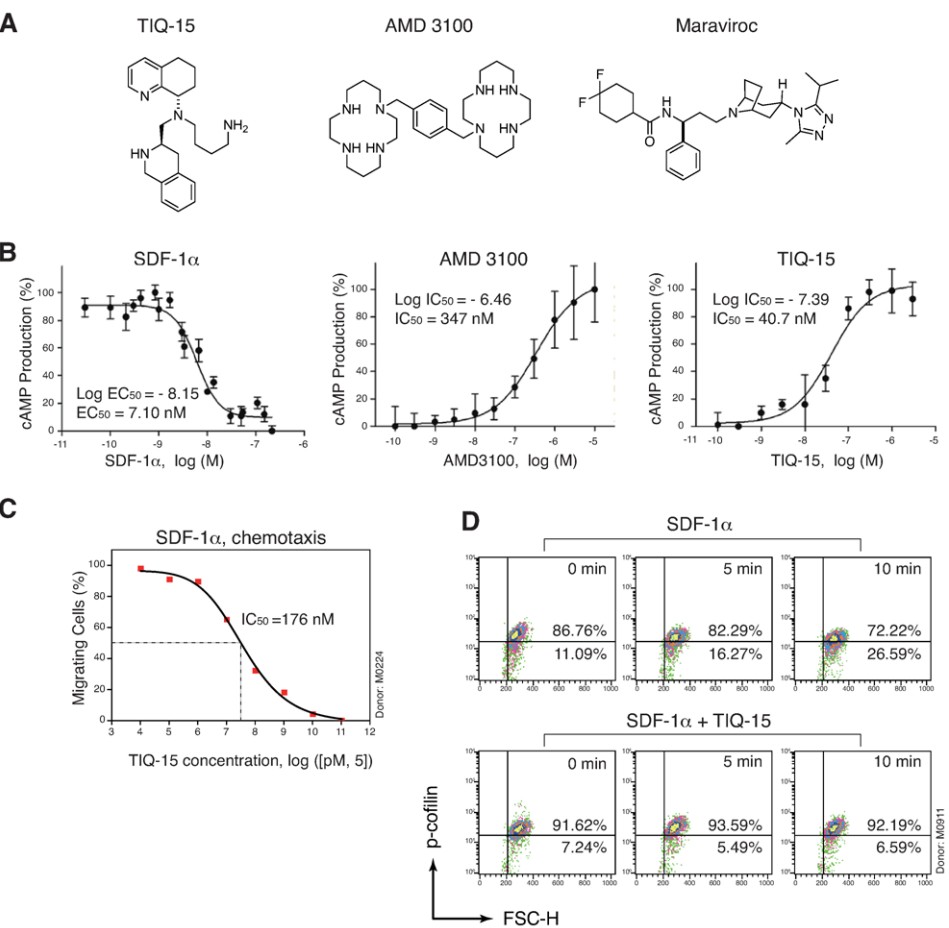

**Fig 1. Characterization of TIQ-15 as a Gαi-based CXCR4 antagonist.** (**A**) Structures of TIQ-15, AMD3100, and maraviroc used in this study. (**B**) SDF-1α inhibits while TIQ-15 or AMD3100 promotes SDF-1α/Forskolin-induced cAMP production. CXCR4-Glo cells were stimulated with 1 μM forskolin to initiate cAMP production. Cells were also co-treated with 28 nM of SDF-1α or 28 nM SDF-1α plus various dosages of AMD3100 or TIQ-15. Luminescence was measured, and the IC50 values were calculated. (**C**) TIQ-15 inhibits SDF-1α-mediated chemotaxis. Resting CD4+ T cells were pretreated with TIQ-15 from 640 pM to 50 μM (5-fold dilution) or DMSO (1%, control) for 1 hour at 37˚C, and then assayed for migration towards SDF-1α (50 nM) in trans-well assays. Results are expressed as the relative percentage of migrating cells. (**D**) TIQ-15 inhibits SDF-1α-mediated cofilin activation. Resting CD4+ T cells were treated with TIQ-15 (10 μM) or DMSO (1%, control) for 1 hour at 37˚C prior to stimulation with SDF-1α (50 ng/ml). Cells were fixed, stained with an anti-p-cofilin antibody, and then analyzed with flow cytometer.

activation of the PKA enzyme [43]. Given that SDF-1α triggers rapid actin rearrangement in resting CD4 T cells, we asked whether TIQ-15 can inhibit SDF-1α/CXCR4-transduced chemotactic signaling in resting T cells. We first performed a TIQ-15 dosage-dependent assay for the inhibition of SDF-1α/CXCR4-mediated chemotaxis. Resting T cells were pretreated with TIQ-15 or DMSO, and then treated with 50 nM SDF-1α. Chemotaxis of CD4 T cells was measured in a trans-well assay. We observed a dosage-dependent inhibition of T cell migration, with an IC50 of 176 nM (**Fig 1C**).

A major signaling pathway mediated by CXCR4 is the Rac1/Rho/cdc42-PAK-LIMK-cofilin pathway [22,23]. We further tested whether inhibition of chemotaxis by TIQ-15 acts through blocking SDF-1α /CXCR4 interaction leading to the inhibition of cofilin activation. Resting CD4 T cells were pre-treated with TIQ-15 for 1 hour. Following TIQ-15 pre-treatment, cells were treated with 50 ng/mL of SDF-1α for a time course of 5 to 10 minutes, and cofilin

activation was measured by intracellular staining of phosphorylated cofilin (p-cofilin). SDF-1α induced a measurable increase in cofilin dephosphorylation, while TIQ-15 blocked cofilin dephosphorylation (**Fig 1D**). Thus, TIQ-15 inhibits SDF-1α /CXCR4-mediated cofilin activity, further supporting its interaction with CXCR4 and acting as an antagonist.

## TIQ-15 interferes with HIV NefM1 peptide-CXCR4 interaction

It has been shown that the HIV Nef protein can induce CD4 T cell apoptosis [44], and a Nef peptide, NefM1, exhibits a strong interaction with CXCR4 and induces Jurkat cell membrane depolarization, caspase-9 activation, DNA laddering, and cell apoptosis [45]. For additional evidence of TIQ-15's specificity in interacting with CXCR4, we examined TIQ-15's ability to block NefM1/CXCR4-mediated apoptosis in Jurkat T cells. Surprisingly, we observed strong inhibition of membrane depolarization induced by NefM1 with a TIQ-15 $IC_{50}$ of 1 nM (**Fig 2A–2E**). To benchmark this result, and to determine the mechanistic coupling to the CXCR4 receptor, we also tested the CXCR4 antagonist AMD3100 and the CCR5 antagonist maraviroc. AMD3100 inhibited NefM1-induced apoptosis with a ~500-fold higher $IC_{50}$ of 474 nM (**Fig 2E**), while maraviroc's selectivity for CCR5 resulted in its failure to block the apoptotic effects (**S2 Fig**). Therefore, our results further validate the specific interaction of TIQ-15 with CXCR4 in blocking NefM1-CXCR4 interaction.

## TIQ-15 inhibits HIV-1 infection of human CD4 T cells

To quantify the anti-HIV activity of TIQ-15, we used a highly stringent HIV Rev-dependent indicator cell line, Rev-CEM-GFP-Luc, in which reporter expression is strictly dependent on HIV Rev, but not on drug-mediated cellular activities [39,40] (**Fig 3A**). Cells were pretreated with TIQ-15 (10 to 250 μM) or DMSO, and then infected with HIV-1(NL4-3) for 2 hours. Cell-free virus and the drug were washed away after infection, and cells were incubated without TIQ-15 for 3 days. We observed complete inhibition of HIV-1 by TIQ-15 at all three dosages without detectable cytotoxicity (**Fig 3A** and **3C**). To accurately measure the $IC_{50}$, we infected Rev-CEM-GFP-Luc following pretreatment of cells with TIQ-15 (from 10 μM to 25.6 pM, 5-fold dilution). Luciferase activity was measured, and we observed dosage-dependent inhibition of HIV-1 with an $IC_{50}$ of 13 nM (**Fig 3B**). To further determine whether TIQ-15 inhibits HIV-1 latent infection of blood resting CD4 T cells, we purified human resting T cells from peripheral blood (98% pure), and pretreated cells with TIQ-15, and then infected cells with HIV-1(NL4-3). Following infection and washing, cells were cultured in the absence of TIQ-15 for 5 days, and then activated by CD3/CD28 stimulation. HIV replication was monitored by virion release (p24 ELISA) (**Fig 3D**). We observed a dosage-dependent inhibition. At day 10 post infection, the $IC_{50}$ value was 79 nM in this particular donor. To confirm this result, we repeated the experiment using resting CD4 T cells from 2 additional donors, and observed similar dosage-dependent inhibition of HIV infection (**Fig 3D**). Nevertheless, there were apparent donor-dependent variations with $IC_{50}$ varying between 34 nM and 107 nM. We also measured effects of TIQ-15 for inhibiting T cell activation. There was no detectable inhibition of T cell activation at all the tested dosages (**S3 Fig**), demonstrating that the inhibition of HIV did not result from inhibition of T cell activities.

## TIQ-15 inhibits viral entry into primary CD4 T cells

We further investigated possible mechanisms of viral inhibition. First, we measured whether TIQ-15 affects the expression of CD4 or CXCR4 on resting CD4 T cells. As shown in **Fig 4A**, we observed down modulation of CXCR4 following a high dosage of TIQ-15 treatment (10 μM), while there was only a minor change in CD4 (**Fig 4B**). We also confirmed CXCR4

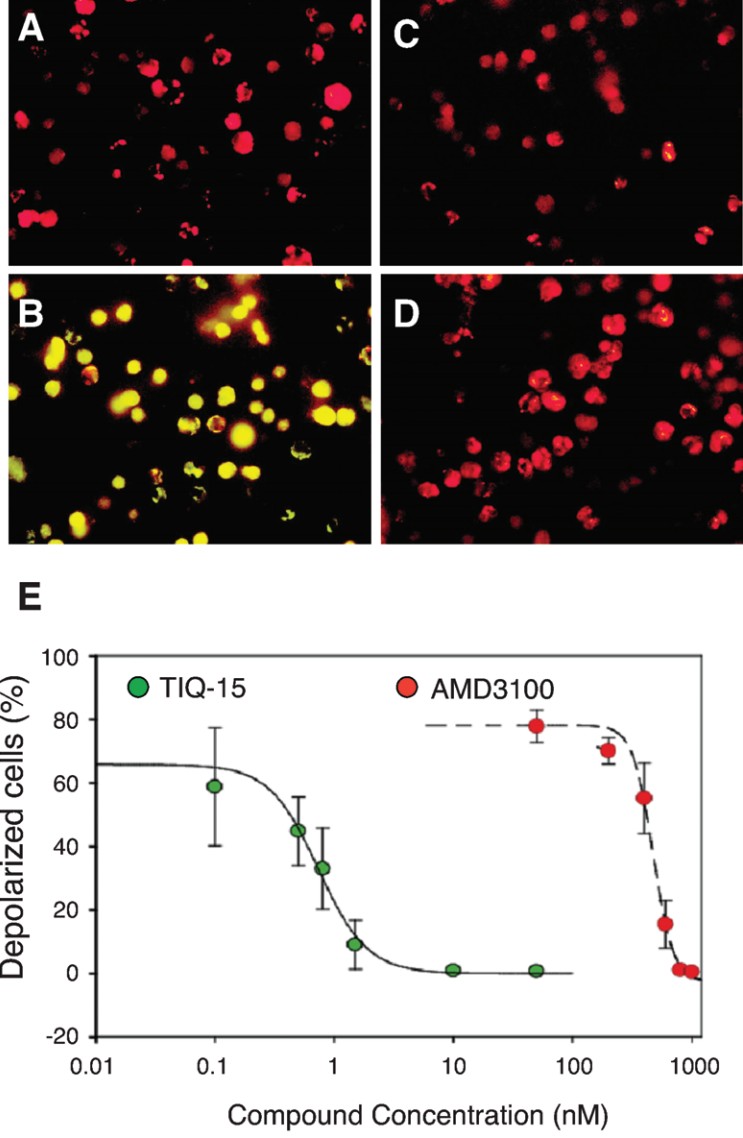

**Fig 2. TIQ-15 inhibits NefM1-CXCR4-mediated cell depolarization in Jurkat T cells.** Jurkat cells were treated with 10 ng/ml of the NefM1 peptide (TNAACAWLEAQ) and different doses of TIQ-15 or AMD3100 for 24 hours. (**A**) Untreated; (**B**) NefM1 treated; (**C**) NefM1 + AMD3100 (1000 nM); (**D**) NefM1 + TIQ-15 (50 nM). Cells were stained with JC-1 and imaged with epi-fluorescent microscopy. Image processing was conducted with Image-Pro 2.0. (**E**) IC50 determinations for TIQ-15 (1 nM) and AMD3100 (474 nM) inhibition of NefM1 depolarization (Numerical and graphical data analyses were conducted using SigmaPlot 10).

downregulation on A3R5 CD4 T cells by treating cells with different dosages of TIQ-15 (from 1 nM to 10 μM), and observed dosage-dependent downregulation of CXCR4 (**Fig 4A**). In contrast, TIQ-15 does not downregulate CCR5 even at 50 μM (**Fig 4G**). The results were consistent with the selectivity of TIQ-15 as a CXCR4 antagonist. To assess the ability of TIQ-15 to block gp120-CXCR4 interaction, we pseudotyped HIV-1 with the vesicular stomatitis virus glycoprotein (VSV-G), which bypasses the CD4/CXCR4 receptors for viral entry via endocytosis. As shown in **Fig 4C**, TIQ-15 did not inhibit VSV-G-pseudotyped HIV-1 infection. On the other hand, it completely blocked HIV-1(NL4-3) infection at the same dosage. These results

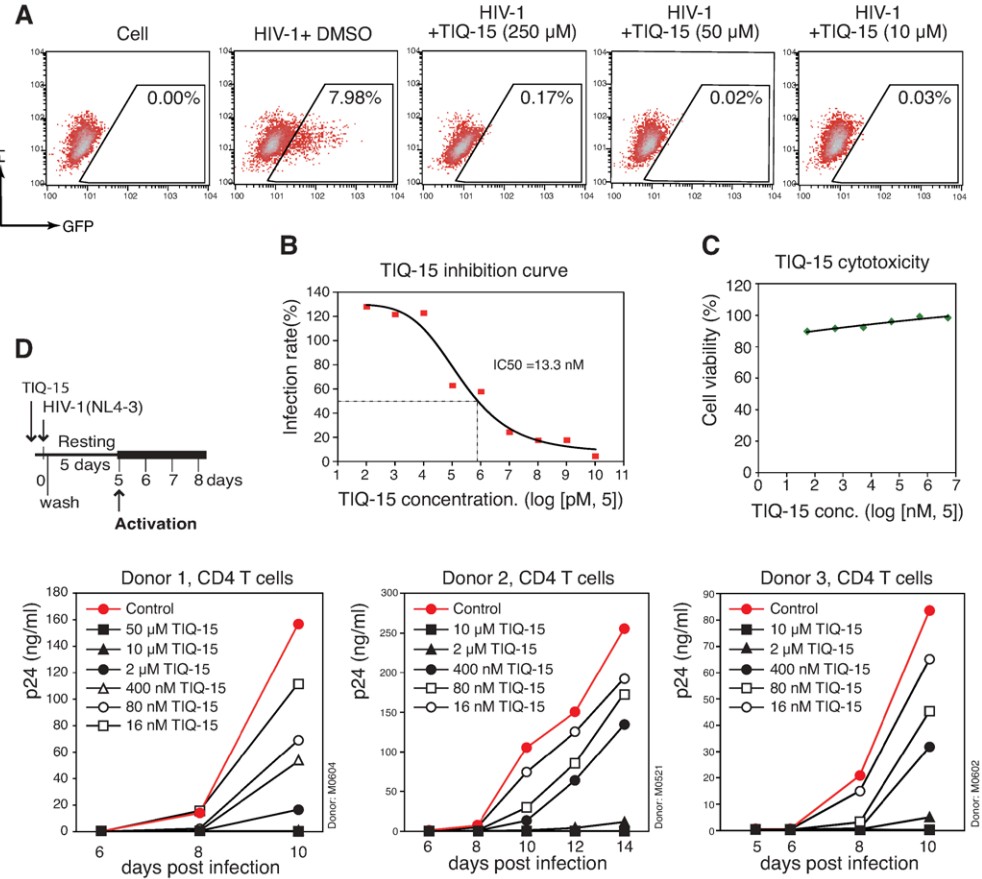

**Fig 3. TIQ-15 inhibits HIV infection of primary and transformed CD4 T cells.** (**A**) TIQ-15 inhibits HIV infection of Rev-CEM-GFP-Luc reporter cell. Cells were pretreated with TIQ-15 at 3 different dosages for 1 hour at 37˚C, and then treated with HIV-1(NL4-3) for 2 hours. Cells were washed and cultured in the absence of TIQ-15 for 48–72 hours, analyzed with flow cytometry. Propidium iodide (PI) was used to ensure GFP quantification only in the viable cell population. (**B**) Quantification of anti-HIV potency. Rev-CEM-GFP-Luc cells were pretreated with TIQ-15 at different dosages (from 10 μM to 25.6 pM, 5-fold dilution) or DMSO for 1 hour at 37˚C, and then treated with HIV-1 (NL4-3) for 2 hours. Cells were washed and cultured in the absence of TIQ-15 for 72 hours, and quantified for luciferase activity. (**C**) Quantification of TIQ-15 cytotoxicity. Rev-CEM-GFP-Luc cells were similarly treated with TIQ-15 from 50 μM to 16 nM (5-fold dilution) or DMSO for 1 hour at 37˚C, washed, cultured for 72 hours, analyzed with flow cytometry for drug cytotoxicity. Results are expressed as the relative percentage of viable cells. (**D**) TIQ-15 inhibits HIV infection of blood resting CD4 T cells. Cells from 3 donors were treated with TIQ-15 at different dosages or with DMSO for 1 hour at 37˚C, and then infected with HIV-1(NL4-3) for 2 hours. Cells were washed and cultured (with IL-7, 1 ng/mL) in the absence of TIQ-15 for 5 days, and then activated with anti-CD3/CD28 magnetic beads (4 beads per cell). Viral replication was measured by HIV-1 p24 release.

demonstrate that TIQ-15 blocks HIV infection specifically at the stage of gp120-CXCR4 interaction rather than any post-entry step or CXCR4-independent entry. To further confirm that TIQ-15 indeed inhibited HIV at the entry step, we performed a BlaM-Vpr-based viral entry assay, and observed complete inhibition of viral entry by TIQ-15, similar to the result observed using AMD3100 (**Fig 4D**). In addition, a time course of viral DNA synthesis, shown in **Fig 4E**, confirmed a lack of viral DNA in TIQ-15-treated cells, consistent with the inhibition of viral entry. Together, our results suggest that TIQ-15 is a CXCR4 antagonist that blocks CXCR4-tropic HIV-1 entry. To further differentiate effects of TIQ-15 on the CXCR4- versus CCR5-mediated viral entry, we performed an experiment with a CCR5-tropic virus (**Fig 4F**). We tested TIQ-15 inhibition of HIV-1(AD8) (R5 virus) on Rev-A3R5-GFP reporter T cells.

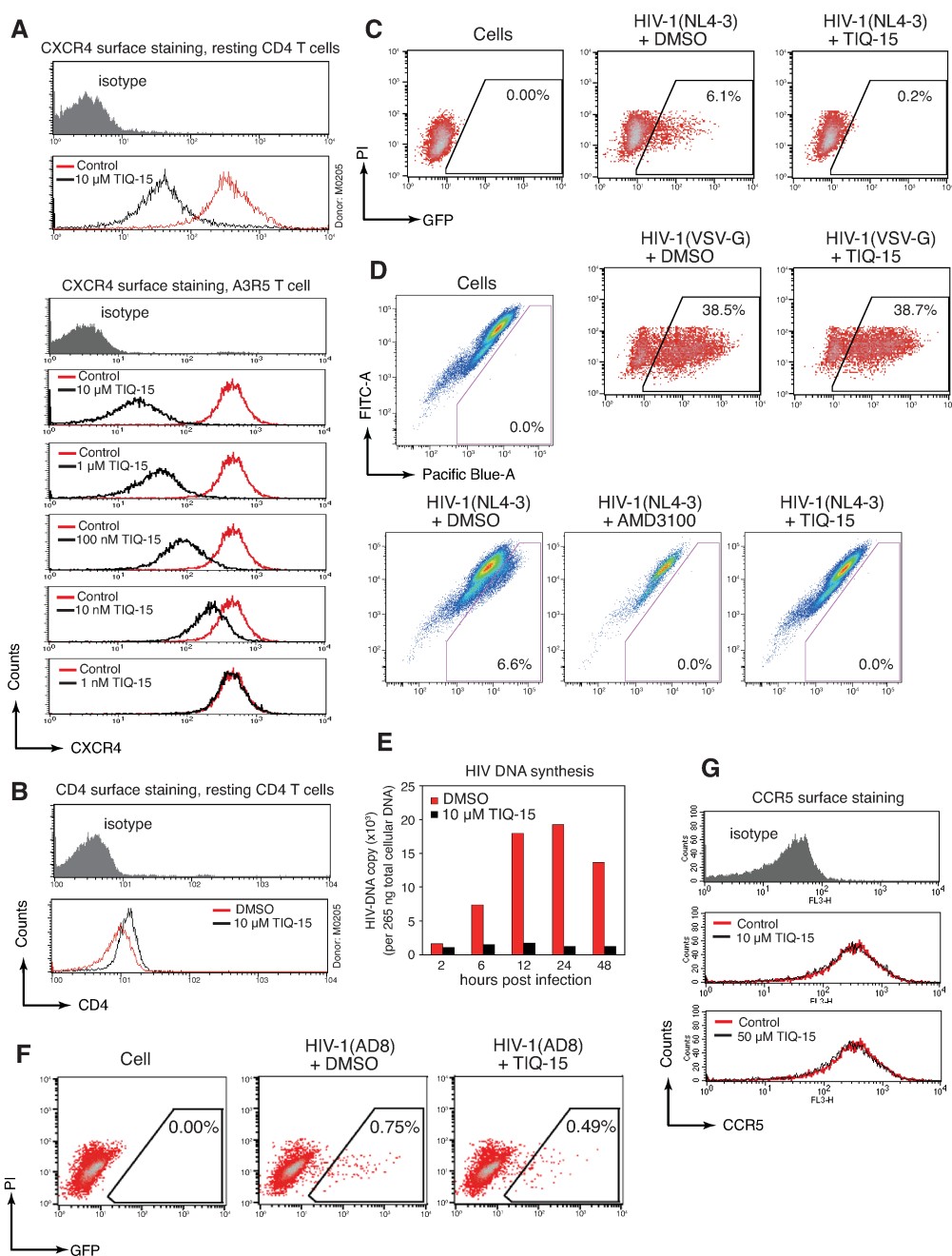

**Fig 4. TIQ-15 blocks HIV entry and downregulates CXCR4.** (**A**) TIQ-15 dosage-dependent downregulation of CXCR4. Resting CD4 T cells were treated with TIQ-15 (10 μM) or DMSO control for 1 hour at 37˚C. Cells were stained with a PE/Cy5-labeling anti-CXCR4 antibody and analyzed with flow cytometer (top panel). A3R5 CD4 T cells were treated with different doses of TIQ-15 (10 μM to 1 nM) and similarly analyzed (lower panel). (**B**) TIQ-15 does not affect CD4 surface density. TIQ-15-treated resting CD4 T cells were stained with a FITC-labeling anti-CD4 antibody. (**C**) TIQ-15 does not inhibit VSV-G pseudotyped HIV-1. Rev-CEM-GFP-Luc cells were treated with TIQ-15 (50 μM) or DMSO for 1 hour at 37˚C, and then infected with HIV-1(VSV-G) or HIV-1(NL4-3) for 2 hours. Cells were washed and cultured in the absence of TIQ-15 for 48 to 72 hours, and analyzed with flow cytometry. (**D**) TIQ-15 blocks HIV entry. CEM-SS cells were treated with TIQ-15 (10 μM), AMD3100 (10 μM) or DMSO for 1 hour at 37˚C, and then infected with BlaM-Vpr containing HIV-1(NL4-3) for 4 hours. Cells were analyzed with a BlaM-Vpr-based entry assay. (**E**) Lack of HIV DNA in TIQ-15-treated and HIV-infected T cells. CEM-SS cells were treated with TIQ-15 (10 μM) or DMSO for 1 hour at 37˚C, and then infected with a single cycle HIV-1(NL4-3)(Env) that was pseudotyped with HIV-1 gp160. Following infection, viral DNA synthesis was followed in a time course and quantified using real-time PCR. (**F**) TIQ-15 moderately inhibits low-dose HIV-1(AD8) infection of Rev-

A3R5-GFP reporter cells. Cells were treated with 10 µM of TIQ-15 for 1 hour, infected with HIV-1(AD8) for two hours. Cells were washed and cultured in the absence of TIQ-15 for 48 hours, and analyzed with flow cytometry. (**G**) TIQ-15 does not affect CCR5 surface density. A3R5 CD4 T cells were treated with TIQ-15 (10 µM and 50 µM) or control for 1 hour at 37˚C. Cells were stained with a PE/Cy5-labeling anti-CCR5 antibody and analyzed with flow cytometer.

This cell line contains both CXCR4 and CCR5 and permits low-level R5 viral infection. The cells were pretreated with 10 µM of TIQ-15 for 1 hour, infected with low-dose HIV-1(AD8), washed, and then cultured for 48 hours in the absence of TIQ-15. Infected cells were analyzed by flow cytometry. As shown in **Fig 4F**, in contrast to the complete inhibition of the X4 virus (**Fig 3A**), we observed only moderate (35%) inhibition of the R5 virus by 10 µM TIQ-15 at the low viral inoculum (infecting 0.75% cells). We repeated the experiment using higher viral inocula (infecting 4% and 14%, respectively), and found that at 10 µM TIQ-15, HIV(AD8) replication was minimally reduced (5%), while at 50 µM, HIV(AD8) replication was reduced to around 50% (**S4 Fig**). This moderate inhibition of the R5 virus at high dosages (10–50 µM) was unexpected, and thus, we further quantified the ability of TIQ-15 to downregulate CCR5 and found no CCR5 downregulation even at 50 µM (**Fig 4G**), which is in great contrast to the potent downregulation of CXCR4 by TIQ-15 at 10 µM (**Fig 4A**). In addition, we performed viral entry assay and found that TIQ-15 only moderately inhibited HIV(AD8) entry at 10–50 µM (**S5 Fig**), which is also in great contract to the complete inhibition of the T-tropic HIV (NL4-3) entry by TIQ-15 and AMD3100 (**Fig 4D**). These results are also supported by an independent binding assay involving CCR5 and one of its cognate ligands, macrophage inflammatory protein-1 alpha. In this assay, HEK-293 cells were incubated with both [125]I-MIP-1$\alpha$ and TIQ-15 at a single concentration (100 µM). It was found that TIQ-15 inhibited MIP-1 binding to CCR5 by 58% (**see Discussion**). Taken in aggregate, the experiments with CCR5 indicate weaker interactions between TIQ-15 and CCR5 than TIQ-15 and CXCR4.

## TIQ-15 inhibited a mixed-tropism panel of HIV clinical isolates

We tested TIQ-15 for inhibition of a panel of clinical HIV isolates. Seven HIV-1 isolates from HIV-1 group M envelope subtypes (A-G) were selected, including two X4-tropic isolates, three R5-tropic isolates, and two X4R5 dual-tropic isolates. TIQ-15 was tested against each of these isolates in the infection of PBMCs (**Fig 5A** and **5B** and **Table 1**). TIQ-15 inhibited HIV-1 differently depending on viral tropisms, with the expected strongest inhibition against the X4-tropic isolates (92UG046 and CMU02, $IC_{50}$ values of 0.78 and 2.16 nM), and one dual-tropic isolate (93BR020, $IC_{50}$ of 1.08 nM) (**Fig 5A** and **Table 1**). We also observed modest but significant inhibition against all 3 R5-tropic isolates (91US001, 98US-MSC5016, and JV1083, $IC_{50}$ values of 820, 587, and 1,320 nM), and one dual-tropic isolate (00KE-KER2008, $IC_{50}$ of 71.4 nM) (**Fig 5B** and **Table 1**). In addition, the $IC_{90}$ values reflect high levels of inhibition for the two X4-tropic, and one dual-CXCR4 preferring isolate in the range of 4.97–20.2 nM, while the activity of TIQ-15 against R5-tropic viruses is much lower ($IC_{90}$ range = 3.8–8 µM) and likely outside the range needed for maintaining effective drug concentrations against R5-tropic viruses as a single agent. Furthermore, the cytotoxicity levels were measured and the $TC_{50}$ was found to be quite high in µM range ($TC_{50}$ = 36 µM). This low toxicity profile provides a large therapeutic index (TI >10,000) for the X4 preferring isolates. However, for the R5-tropic isolates, this index is much lower (~40-fold), which indicates that the R5-based activity of TIQ-15 is not useful as a stand-alone R5 virus inhibitor. However, this moderate anti-R5 virus activity of TIQ-15, although not sufficiently potent for R5 viruses when used alone, may make it amenable to be co-administered as a synergistic drug with CCR5 inhibitors such as maraviroc. Several dual CXCR4-CCR5 chemokine agents have been disclosed in the literature; however, to

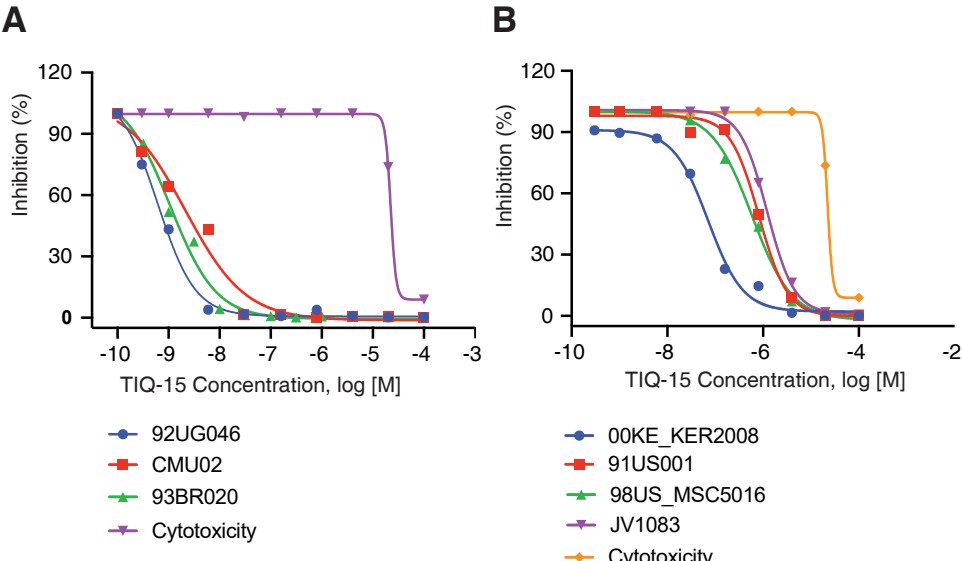

**Fig 5. TIQ-15 inhibits HIV clinical isolates of X4, R5, and mixed tropisms.** Human PBMCs were cultured, pre-treated with TIQ-15 at various dosages, and then infected with HIV for 7 days. Cell-free supernatants were collected and analyzed for reverse transcriptase activity. TIQ-15 potently inhibits infection of PBMCs by two X4- and one dual-tropic viruses (**A**). TIQ-15 is moderately active against three R5-tropic and one dual-tropic viruses (**B**). Compound cytotoxicity was measured by MTS assays.

our knowledge, this is the first report of a potent CXCR4-based HIV entry inhibitor with significantly synergistic anti-R5 virus activity.

## TIQ-15 is synergistic with maraviroc against the R5-tropic HIV-1 strain Ba-L

In light of our unexpected finding that TIQ-15 exhibits very potent inhibition of X4-based HIV entry, along with a weaker but significant ability to block infection by R5-tropic viruses, we were interested in studying its anti-HIV activity when co-administered with maraviroc. A two-drug combination study was performed with TIQ-15 and maraviroc using the Prichard and Shipman MacSynergy II three dimensional model for statistical evaluation [46]. This analytical technique combined with *in vitro* testing data is accepted as a surrogate for the determination of potential anti-retroviral drug combination effects. The combination anti-viral assays were performed with MAGI-CCR5 cells, utilizing either HIV-1(IIIB) (X4-tropic) or HIV-1

**Table 1. Inhibition of clinical isolates of HIV-1 in PBMCs by TIQ-15.**

| HIV-1 clinical isolate | Envelope subtype | Tropism | IC$_{50}$ (nM)[a] | IC$_{90}$ (nM)[a] |
|---|---|---|---|---|
| 92UG046 | D | CXCR4 | 0.78 | 4.97 |
| CMU02 | EA | CXCR4 | 2.16 | 20.2 |
| 93BR020 | F | Dual/Mixed | 1.08 | 8.12 |
| 00KE-KER2008 | A | Dual/Mixed | 71.4 | 357 |
| 91US001 | B | CCR5 | 820 | 3,840 |
| 98US-MSC5016 | C | CCR5 | 587 | 3,510 |
| JV1083 | G | CCR5 | 1,320 | 8,020 |

[a]Cytotoxicity: TC$_{50}$ = 36,000 nM.

**Table 2. Determination of the anti-HIV-1 synergy for combination doses of TIQ-15 and Maraviroc.**

| HIV-1 virus/ test | Mean antiviral efficacy vs. HIV-1 synergy/antagonism volume $(nMmM^2\%)$ | Interpretation/result |
|---|---|---|
| IIIB (X4) | 36.8/-0.26 | Additive/Not synergistic |
| Ba-L (R5) | 149/-0.14 | Highly synergistic |
| Cytotoxicity | 0/-14.0 | Additive/Not synergistic |

(Ba-L) (R5-tropic). A broad range of five concentrations of maraviroc were tested, in all possible combinations, with eight concentrations of TIQ-15, which were selected to provide a comprehensive dose-response curve for both test articles. The data was then analyzed according to the method of Prichard and Shipman [46] using the MacSynergy II program for data analysis and statistical evaluation. The program calculates the theoretical additive interactions of the drugs based on the Bliss independence mathematical definition of expected effects for drug-drug interactions. The Bliss independence model, also known as Dual-Site (DS) model, is based on statistical probability, and assumes that the drugs act independently to affect virus replication. Theoretical additive interactions were calculated from the dose response curves of each individual drug. This calculated additive surface, which represents predicted or additive interactions, was then subtracted from the experimentally determined dose-response surface to reveal regions of non-additive activity (**S7 Fig**). The results of the drug combination analysis are shown in **Table 2** for HIV-1(IIIB) and HIV-1(Ba-L). For the X4-tropic HIV-1(IIIB), three-dimensional evaluation revealed some small additive effect with no observable antagonism or reduction of anti-viral activity (**Table 2**). However, results from the R5-tropic HIV-1(Ba-L) revealed a more pronounced effect (**Table 2**). The synergy volume was about 4–5 fold higher with no concurrent measurable antagonism. This synergistic effect seen with the R5-tropic virus is consistent with the weak concurrent anti-R5-tropic activity of TIQ-15 observed in the HIV-1 clinical isolate panel (**Fig 5B**). The overall results from this study suggest a potential synergistic inhibition of R5-tropic viruses by TIQ-15 and maraviroc.

## Discussion

In this study, we demonstrate that TIQ-15 is a novel tetrahydroisoquinolines-based CXCR4 antagonist that restricts Gαi-based SDF-1 signaling, thereby blocking HIV-1 entry and infection (**Table 3**). TIQ-15 exhibited anti-viral activity with potent effects in different cell-based assays. It is a potent inhibitor of T-tropic HIV-1 infection both in HIV Rev-dependent indicator T cells ($IC_{50}$ = 13 nM), PBMCs, and blood resting CD4 T cells, and was non-toxic to these cells at concentrations up to 50 μM. Our results from viral entry assay and VSV-G pseudo-typed virus infection confirmed that the inhibition of CXCR4-mediated viral entry is the primary mechanism of action of TIQ-15. Moreover, CXCR4-based studies confirm that the compound blocks SDF-1α/CXCR4 signaling and cofilin activation through the Gαi pathway. The pharmacology of TIQ-15 as a CXCR4 antagonist has also been tested previously (**Table 3**). TIQ-15 inhibits the function of the receptor in many different assays including calcium flux ($IC_{50}$ = 5–10 nM), $^{125}$I-SDF binding ($IC_{50}$ = 112 nM), and beta-arrestin recruitment inhibition ($IC_{50}$ = 19 nM). Results from this study (cAMP, $IC_{50}$ = 41 nM; chemotaxis, $IC_{50}$ = 176 nM) provides similar outcomes showing that TIQ-15 binds to and blocks the natural ligand SDF-1 from acting on the receptor in a similar concentration range. Furthermore, TIQ-15 treatment in a calcium flux assay absent of SDF-1 shows no effects, indicating that TIQ-15 is not a CXCR4 agonist. The data showing blockade of SDF-1 receptor internalization via the

**Table 3. Summary of results on TIQ-15.**

| Bio-assay | Result |
|---|---|
| Inhibition of CXCR4 cAMP regulation | $IC_{50}$ = 41 nM |
| Inhibition of SDF-1α-inuced chemotaxis of CD4 cells | $IC_{50}$ = 176 nM |
| Inhibition of $^{125}$I-MIP1α binding in HEK-293T cells | 58% @ 100 μM |
| Inhibition of HIV-1 infection of Rev-CEM cells | $IC_{50}$ = 13 nM |
| Cytotoxicity in Rev-CEM cells | $IC_{50}$ > 250,000 nM |
| Inhibition of NefM1 Induced Jurkat Depolarization | $IC_{50}$ = 1 nM |
| Inhibition of HIV-1 (X4, R5, and X4R5) infection in PBMCs<br>HIV-1 (X4)<br>HIV-1 (R5) | $IC_{50}$ range, 1 nM (X4) to 1,320 nM (R5)<br>$IC_{50}$ = 2 nM (average)<br>$IC_{50}$ = 897 nM (average) |
| Cytotoxicity in PBMCs | $TC_{50}$ = 36,000 nM |
| *In Vitro* HIV-1 therapeutic index in PBMCs<br>X4 envelope HIV-1<br>R5 envelope HIV-1 | 18,000 (average)<br>40 (average) |
| *In Vitro* CXCR4/SDF-1α/HIV-1 selectivity index in PBMCs<br>($IC_{50}$ cell migration/$IC_{50}$ X4 HIV-1) | 88 (average) |

beta-arrestin pathway is similar in potency to the other assays of blocking receptor signaling (Calcium flux, cAMP; 19 nM versus 10 and 41 nM), indicating that TIQ-15 blocks receptor internalization via affecting a signaling pathway. Additionally, surface CXCR4 staining in this study indicates that TIQ-15 itself downmodulated CXCR4 in a dosage-dependent manner, which decreased CXCR4 density around 50% at 10 μM, but minimally downmodulated CXCR4 at dosages below 10 nM (**Fig 4A**). Given that TIQ-15 inhibits X4-tropic HIV with an $IC_{50}$ of 13.3 nM (**Fig 3B**), these results do not support CXCR4 downregulation as the primary antiviral mechanism of TIQ-15. It is likely that at clinically effective concentrations of TIQ-15, the CXCR4 density is minimally affected.

TIQ-15-mediated downmodulation of CXCR4 at high dosages could result from a TIQ-15 promoted receptor internalization, dimerization, or receptor cycling, but this occurs at a much higher concentration than inhibition of signaling (10–50 nM versus 10 μM), and thus, is apart from and independent of blockade of receptor signaling. CXCR4 downregulation is more likely a unique and under reported effect rather than a rebuke to the characterization of TIQ-15 as a CXCR4 receptor antagonist. For example, AMD3100 has been reported to cause CXCR4 receptor tolerance in certain cell types [47]. There could be a similar effect with TIQ-15 in our study. In summary, our work provides a mechanistic understanding of TIQ-15 as a CXCR4 allosteric antagonist with the ability to block HIV-1 entry and HIV-1/CXCR4 signaling.

TIQ-15 can also inhibit Nef-induced apoptosis. In similar studies, AMD3100 was 100-fold less active, while maraviroc was completely inactive (**Figs 2E** and **S2**), which further emphasizes the involvement of CXCR4 in Nef induced-apoptosis and the specific interaction of TIQ-15 with CXCR4.

As an HIV entry inhibitor, TIQ-15 was shown to inhibit mainly X4-tropic HIV-1 strains, with $IC_{50}$ and $IC_{90}$ values observed in the lower nano molar range, which is indicative of low plasma levels needed for anti-viral effects. Furthermore, pre-treatment of HIV infected cells with TIQ-15 completely inhibited infection, suggesting the possibility for its use as a preventative agent (PrEP). Given that the most commonly transmitted HIV-1 variants are M-tropic, and utilize the CCR5 receptor [18], the anti-viral activity observed of TIQ-15 against both dual/mixed, and M-tropic HIV isolates is significant. Although weaker in potency (500–1,000 nM), the dual-tropism activity of TIQ-15 provides insights into its potential as a dual target entry inhibitor that would be tropism independent. Other characterizations of TIQ-15's

interactions with CCR5 reveal that the compound has weaker potency against the natural ligand MIP-1α (58% inhibition of binding, **Table 3**) and doesn't affect CCR5 receptor populations (**Fig 4G**). However, although one compound (AMD3451) has been reported with weak (μM) activity against both CXCR4 and CCR5 strains, TIQ-15 represents the next step in this evolution as it possesses more potent T-tropic activity while exhibiting milder M-tropic anti-HIV-1 activity [48]. Another compound series (Pyrrolo-piperazines) was recently also found to possess dual-tropic anti-HIV-1 activity, but also with concomitant reverse transcriptase enzyme inhibition [49]. Unlike TIQ-15, these two compound series lack the sufficient potency against HIV to be utilized as HIV agents. To our knowledge, the dual-tropic profile result for TIQ-15 has not been reported for any CXCR4 based HIV entry inhibitors. In consideration of early intervention, CXCR4 seems to be a less attractive target for HIV treatment, since T-tropic strains only appear at later stages of HIV infection in around 50% of infected individuals [15–17]. However, the unique mechanism of action of TIQ-15 can hardly be overlooked, as this compound contains weaker alternate tropism activity that boosts the R5 activity of maraviroc.

This high potency of TIQ-15 against X4-tropic viruses is significant since the emergence of X4-tropic strains is associated with a subsequent rapid CD4 T cell decline, and an accelerated disease progression [50,51]. The rapid CD4+ T cell loss after the emergence of X4-tropic strains is due to a large proportion of CXCR4 expressing CD4 T cells. Compared to CCR5 expression (15–35%), CXCR4 is expressed on 90% of CD4 T cells, especially on naïve CD4+ T cells, the largest population of CD4 T cells in the human body [20,52]. Moreover, the emergence of X4-tropic variants was shown to develop CCR5 antagonist resistance in anti-HIV therapy [35]. Hence, the CXCR4 antagonist, TIQ-15, could find utility in all phases of HIV infection, first having synergistic activity with a CCR5 agent, then providing protection against developing T-tropic strains. Hence, TIQ-15 has the potential to be a promising complement to existing antiviral therapies for complete viral suppression.

SDF-1α acts as an endogenous inhibitor of HIV-1 by blocking common binding sites that are required for gp120/CXCR4 interaction or by inducing receptor internalization [53,54]. TIQ-15 is a unique allosteric CXCR4 receptor antagonist that likely binds to the CVX-15 peptide site [55]. Dosage-dependent inhibition results reported above for SDF-1α-mediated chemotaxis confirmed TIQ-15's specificity for binding CXCR4. Inhibition of SDF-1α binding to CXCR4 by TIQ-15 is necessary for avoiding cross reaction and immune dysregulation by non-specific binding. The results of SDF-1α chemotaxis inhibition also indicated the interference of SDF-1α/CXCR4 signaling axis by TIQ-15. Adverse effects due to inhibition of SDF-1α/CXCR4 interaction have been observed in SDF-1α [56] or CXCR4 [57] knockout in mice, resulting in a series of significant defects on development. However, it should be noted that in another test of TIQ-15, acute toxicity was not observed in adult mice [38]. Furthermore, CXCR4 antagonists are being administered to healthy human volunteers, and HIV infected patients with no reported clinical adverse effects, hence, those potential side effects are unlikely to become an issue. Approval of AMD3100 for hematopoietic stem cell mobilization indicates that at least short-term interference of the SDF-1α/CXCR4 axis is safe in humans.

The TIQ-15-mediated inhibition of SDF-1α-induced cofilin activation are consistent with a demonstrated role of early actin dynamics in HIV latent infection of blood resting CD4 T cells [22,25,26,28]. In HIV infection, it has been recently shown that chronic HIV infection in patients causes hyperactivation of cofilin in their blood CD4 T cells that correlates with the degree of T cell damage, and inversely correlates with T cell recovery from ART [25]. Currently, there is no drug treatment for HIV-mediated cofilin hyperactivation. TIQ-15 was shown to bind to CXCR4 in a specific and reversible manner. In addition, drug-mediated interference or diversion of HIV-mediated signaling to actin activity is achievable. Therefore, based on the selectivity and favorable ADME properties of TIQ-15, it is predicted to be a

potential CXCR4 antagonist that could also be used to treat HIV-mediated cofilin hyperactivation.

In conclusion, we have shown that the novel CXCR4 antagonist, TIQ-15, has potent inhibitory activity against X4-tropic HIV-1 via an entry inhibition mechanism. This small molecule exhibits new and unique anti-HIV-1 properties, blocking infection both in Rev-dependent indicator T cells and human primary resting CD4+ T cells without detectable cytotoxic effects. TIQ-15 also blocks infection of multi-tropic virus populations in PBMC, and has the ability to act synergistically with maraviroc against R5-tropic HIV strains. Furthermore, the compound was shown to block CXCR4-based signaling in a specific and reversible manner propelling its potential utility in the treatment of immune dysfunctions caused by HIV and cancers. TIQ-15 also has favorable ADME properties that make it a promising drug candidate (38).

## Materials and methods

### Ethics statement

All protocols involving human subjects were reviewed and approved by the George Mason University Institutional Review Board (IRB). Formal written consent was obtained.

### Cells, viruses and HIV infection

Peripheral resting CD4 T cells were purified from peripheral blood by two rounds of negative selection as previously described [22]. Virus stocks of HIV-1(NL4-3), BlaM-Vpr containing HIV-1(NL4-3), and VSV-G-pseudotyped HIV were prepared as described previously [22,58]. Virus titer ($TCID_{50}$) was measured by infection of a Rev-dependent GFP indicator cell line, Rev-A3R5-GFP [39,40,59,60]. For HIV infection of Rev-CEM-GFP-Luc cells, cells ($2x10^5/0.1$ mL) were incubated with HIV for 2 hours at 37˚C. For infection of blood resting CD4 T cells, cells ($1x10^6/mL$) were infected with HIV for 2 hours at 37˚C, cultured for 5 days in IL-7 (1 ng/mL) (R&D Systems) without stimulation, and then activated with anti-CD3/CD28 magnetic beads [22]. Viral replication was quantified with p24 ELISA using an in-house ELISA kit. For pretreatment of cells with compounds, unless specified, cells were pretreated with TIQ-15, AMD3100 or Dimethyl sulfoxide (DMSO) (1%, as control) for 1 hour at 37˚C, and then infected with HIV. TIQ-15 was synthesized as previously described by Truax et. al. [38]. Additional description is provided in Supporting Information (**S1 Appendix**).

### Infection of PBMCs by HIV clinical isolates and TIQ-15 inhibition

Fresh human PBMCs (Biological Specialty Corporation) ($1–2 x 10^6$ cells/mL) were cultured in 4 μg/mL Phytohemagglutinin (PHA) (Sigma) and 20 U/mL recombinant human IL-2 (R&D Systems) for 48–72 hours at 37˚C. Cells were pretreated with TIQ-15 indicated dosages and then infected with HIV isolates (MOI, 0.1) for seven days. Viral replication was quantified by reverse transcriptase assay, and compound cytotoxicity was measured by MTS assay. HIV-1 isolates 92UG046, CMU02, 93BR020, 00KE-KER2008, 91US001, 98US-MSC5016, and JV1083 were obtained from the NIH HIV Reagent Program. Additional description is provided in Supporting Information (**S1 Appendix**).

### Synergy studies of TIQ-15 with maraviroc

MAGI-CCR5 cells, HIV-1(IIIB) and HIV-1(Ba-L) were obtained from the NIH HIV Reagent Program. For each assay, a pre-titered aliquot of virus (0.001 $TCID_{50}$/cell) was used in a 96 well plate. A checkerboard plate format was used to test five concentrations of drug A (maraviroc) in all possible combinations with eight concentrations of drug B (TIQ-15). Combination

antiviral efficacy was evaluated on three identical assay plates (i.e., triplicate measurements) that include cell control wells (cells only) and virus control wells (cells plus virus). Additional description is provided in Supporting Information (S1 Appendix).

Description of other materials and methods is provided in Supporting Information (S1 Appendix).

## Supporting information

**S1 Fig. Screening and quantification of anti-HIV activity of TIQ compounds.** Rev-CEM-GFP/Luc reporter cell were pretreated with each individual TIQ compounds (10 μM) for 1 hour at 37˚C, and then treated with HIV-1(NL4-3) for 2 hours. Cells were washed and cultured in the absence of TIQ for 48–72 hours, analyzed with flow cytometry. Propidium iodide (PI) was used to ensure GFP quantification only in the viable cell population. (EPS)

**S2 Fig. Maraviroc and AZT do not inhibit NefM1-CXCR4-mediated cell depolarization in Jurkat T cells.** Jurkat cells were treated with 10 ng/ml of the NefM1 peptide (TNAACAW-LEAQ) and different doses of Maraviroc or AZT for 24 hours. (**A**) Maraviroc + NefM1 treated; (**B**) AZT + NefM1 treated. Cells were stained with JC-1 and imaged with epifluorescent microscopy. Image processing was conducted with Image-Pro 2.0. Numerical and graphical data analyses were conducted using SigmaPlot 10. (EPS)

**S3 Fig. TIQ-15 does not inhibit T cell activation.** Resting CD4 T cells were treated with TIQ-15 at different dosages or DMSO (1%. control) for 1 hour at 37˚C, and then treated with HIV-1(NL4-3) virus for 2 hours. Cells were washed and cultured in the absence of TIQ-15 for 5 days, cells were activated with anti CD3/CD28 magnetic beads at day 5 (4 beads per cell), and in day 6, half of the cells in each tube were washed and used for CD69 surface staining to quantify CD69 expression as a marker of T cell activation. (EPS)

**S4 Fig. Quantification of anti-HIV(AD8) activity of TIQ-15.** Rev-A3R5-GFP reporter cell were pretreated with TIQ-15 compound (10 μM and 50 μM) for 1 hour at 37˚C, and then infected with HIV-1(AD8) (**A**, 50 μl; **B**, 500 μl) for 2 hours. Cells were washed and cultured in the absence of TIQ-15 for 48 hours, analyzed with flow cytometry. Propidium iodide (PI) was used to ensure GFP quantification only in the viable cell population (R2, PI-negative). R1 is the PI-positive cells. (EPS)

**S5 Fig. Quantification of TIQ-15 on HIV(AD8) entry.** A3R5 CD4 T cells were treated with TIQ-15 (10 μM and 50 μM) for 1 hour at 37˚C, and then infected with BlaM-Vpr containing HIV-1(AD8) for 4 hours. Cells were analyzed with a BlaM-Vpr-based entry assay. HIV(NL4-3) was used as a control. (EPS)

**S6 Fig. TIQ-15 inhibits HIV(AD8) infection of blood resting CD4 T cells.** Resting CD4 T cells were purified from a donor, treated with TIQ-15 (10μM or 50 μM) for 1 hour at 37˚C, and then infected with HIV-1(AD8) for 2 hours. Cells were washed and cultured in the absence of TIQ-15 for 3 days, and then, activated with anti-CD3/CD28 magnetic beads (4 beads per cell). Viral replication was measured by HIV-1 p24 release. ELISA was performed in duplicate, and the averages of p24 values are shown. (EPS)

**S7 Fig. TIQ-15/Maraviroc Synergy Plot.**
(EPS)

**S1 Data. Source data for tables and graphs in the study.**
(DOCX)

**S1 Appendix. Additional materials and methods.**
(DOCX)

## Acknowledgments

We thank the NIH HIV Reagent Program, Division of AIDS, NIAID, NIH for providing the following reagents: HIV-1 92UG046, HIV-1 93BR020, HIV-1 CMU02, HIV-1 00KE-KER2008, HIV-1 98US_MSC5016; HIV-1 91US001 (GS 004), HIV-1 Jv1083, HIV-1 Ba-L, HIV-1 IIIB, MAGI-CCR5 Cells. The authors wish to thank the discussions and support from Roger H. Miller, Contracting Officer's Representative (COR), Targeted Interventions Branch, Basic Science Program Division of AIDS, NIAID, NIH for the studies of TIQ-15 against the seven tropic HIV isolates and the synergy studies with maraviroc, and also thank Roger G. Ptak, Contract Principal Investigator (PI) for oversight of the work conducted at Southern Research Institute and for review and editing of the manuscript.

## Author Contributions

**Conceptualization:** Lawrence J. Wilson, Yuntao Wu.

**Funding acquisition:** Lawrence J. Wilson, Yuntao Wu.

**Investigation:** Zheng Zhou, Jia Guo, Brian Hetrick, Sameer Tiwari, Amrita Haikerwal, Yang Han, Vincent C. Bond, Ming B. Huang, Marie K. Mankowski, Beth A. Snyder, Priscilla A. Hogan, Savita K. Sharma, Dennis C. Liotta, Terry-Elinor Reid.

**Project administration:** Lawrence J. Wilson, Yuntao Wu.

**Supervision:** Lawrence J. Wilson, Yuntao Wu.

**Validation:** Lawrence J. Wilson, Yuntao Wu.

**Writing – original draft:** Lawrence J. Wilson, Yuntao Wu.

**Writing – review & editing:** Lawrence J. Wilson, Yuntao Wu.

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
