## [Decision Letter · Decision Letter 0]

5 Apr 2024

Dear Dr. Wu,

Thank you very much for submitting your manuscript "Characterization of a CXCR4 antagonist TIQ-15 with dual tropic HIV entry inhibition properties" for consideration at PLOS Pathogens. As with all papers reviewed by the journal, your manuscript was reviewed by members of the editorial board and by several independent reviewers. In light of the reviews (below this email), we would like to invite the resubmission of a significantly-revised version that takes into account the reviewers' comments.

We cannot make any decision about publication until we have seen the revised manuscript and your response to the reviewers' comments. Your revised manuscript is also likely to be sent to reviewers for further evaluation.

Sincerely,

Daniel C. Douek

Academic Editor

PLOS Pathogens

Richard Koup

Section Editor

PLOS Pathogens

Michael Malim

Editor-in-Chief

PLOS Pathogens

orcid.org/0000-0002-7699-2064

Reviewer's Responses to Questions

**Part I - Summary**

Reviewer #1: The investigators here previously identified a chemical inhibitor of CXCR4-tropic HIV-1 that is further characterized in this report. Most of the data here establish the inhibition of CXCRF by TQ-15, including blocking SDF-1 binding, downmodulation of surface CXCR4, and prevention of entry of X4 HIV-1 isolates into cells. These data seem reasonable. However, there is also an effect, albeit weaker, on the replication of R5 virus, and this is then put forth as an important characteristic of the chemical. There is little characterization of how TQ-15 inhibits R5 tropic HIV. Without a mechanism or step in the viral lifecycle identified, this report doesn’t rule out that this is merely an off-target effect or toxicity of the chemical that is not picked up in the cytotoxicity assay. Novelty here relies on the R5 virus mechanism of inhibition.

Reviewer #2: Manuscript by Zhou et al describes studies characterizing small molecule TIQ-15 on suppression of X4-tropic HIV-1 infection. In a previous publication, TIQ-15 was identified as a CXCR4 antagonist and shown to inhibit X4-tropic HIV-1 infection, compete with SDF-1 in a binding assay, and prevent cAMP production. In this manuscript, the actions of TIQ-15 were further studied. Authors found that TIQ-15 inhibits SDF-1a/CXCR4 mediated chemotaxis and T cell migration and interferes with apoptosis induced by NefM1 peptide-CXCR4 interaction. The inhibition of HIV-1 infection was further probed, and they found that 1) CXCR4 is down-regulated upon treating cells with 10 �M TIQ-15, 2) TIQ-15 can inhibit R5- or dual-tropic virus infection albeit at higher IC50 values, and 3) TIQ-15 acts synergistically with Maraviroc to block a R5-tropic virus replication.

The strengths of the manuscript are that TIQ-15 is a lead that targets HIV-1 entry using the CXCR4 coreceptor; currently there is no approved antiviral against this target. Additionally, the synergistic effect with Maraviroc is interesting and intriguing. However, the mechanism(s) by which TIQ-15 inhibits X4- or R5-mediated entry is not addressed. The down regulation of CXCR4 could potentially explain inhibition of X4-tropic virus entry. However, it is unclear how this small molecule inhibits R5-tropic virus infection.

**Part II – Major Issues: Key Experiments Required for Acceptance**

Reviewer #1: Therefore to make claims that this will be a useful dual-tropic inhibitor, more data on inhibition of R5 virus is needed.

Specific criticism:

1) Figure 2 shows an effect of TIQ-15 on T cell polarization. It would be helpful to provide representative images of polarized vs. depolarized cells as assessed by this assay. It was hard to tell from the description what was being assessed, but presumably this is mitochondrial membrane depolarization where depolarization is indicated by a decrease in the red/green fluorescence intensity profile. Please clarify and include in methods.

2) Figure 3 uses X4 tropic virus to examine inhibition of infection of CD4+ T cells. Given the focus on dual-tropic inhibition, this figure would benefit from the use of an R5 tropic strain using the same assay.

3) Figure 4F provides some evidence of inhibition of R5 virus. The control infection rate was quite low, and the effect shown for higher MOI would be useful in judging the actual decrease in infection by R5 virus.

4) Figure 5 is the best example of data showing inhibition of dual tropic and R5 virus along with X4 virus. However, a one-time measurement of RT from supernatants could be complemented by studies to understand where the block in replication is taking place. For R5 virus, is it entry, post-entry, transcription, translation, assembly? The step is not established here, and if it were it would help in understanding if this is a block that is really specific for R5 virus or a general effect on cells that reduces cell health enough to reduce particle production.

Reviewer #2: 1. How does TIQ-15 block X4 virus infection? In a previous study, it was hypothesized that the small molecule binds a pocket in CXCR4. Unfortunately, the current report does not add more insights to the mechanism of action of TIQ-15. A single data point in Fig. 4A shows that at 10 �M, TIQ-15 reduces CXCR4 expression on T cell surface. However, it is unclear that CXCR4 down regulation is responsible for blocking HIV-1 infection at the IC50 or IC90 concentrations. Examining the effect of TIQ-15 treatment on cell surface CXCR4 expression using multiple concentrations and time points may help address whether X4 down regulation is a mechanism of action in inhibiting X4-tropic virus infection.

2. Given its effect on R5-tropic virus, perhaps examining CCR5 expression upon treatment of TIQ-15 will be fruitful.

**Part III – Minor Issues: Editorial and Data Presentation Modifications**

Reviewer #1: (No Response)

Reviewer #2: 1. Results from Fig. 5 and Table 1 are inconsistent. Especially for 00KE_KER2008, Figure 5 listed the IC50 as 71.38 nM whereas Table 1 listed the IC50 as 660 nM.

PLOS authors have the option to publish the peer review history of their article (what does this mean?). If published, this will include your full peer review and any attached files.

Reviewer #1: No

Reviewer #2: No
---

## [Decision Letter · Decision Letter 1]

25 Jul 2024

Dear Dr. Wu,

We are pleased to inform you that your manuscript 'Characterization of a CXCR4 antagonist TIQ-15 with dual tropic HIV entry inhibition properties' has been provisionally accepted for publication in PLOS Pathogens.

Best regards,

Daniel C. Douek

Academic Editor

PLOS Pathogens

Richard Koup

Section Editor

PLOS Pathogens

Michael Malim

Editor-in-Chief

PLOS Pathogens

orcid.org/0000-0002-7699-2064

Reviewer Comments (if any, and for reference):

Reviewer's Responses to Questions

**Part I - Summary**

Reviewer #1: The investigators here previously identified a chemical inhibitor of CXCR4-tropic HIV-1 that is further characterized in this report. Most of the data here establish the inhibition of CXCR4 by TQ-15, including blocking SDF-1 binding, downmodulation of surface CXCR4, and prevention of entry of X4 HIV-1 isolates into cells. In the revised version, they now outline the low-level effect on R5-tropic virus inhibition in a more complete manner. The potential of this to be useful in synergy with a more potent CCRF inhibitor is discussed. The data appear valid and support the arguments put forth by the authors.

Reviewer #2: The revised manuscript added more data in Figure 4, which helped address some of the previous concerns. In general, the manuscript is much improved. There are two minor points that should be addressed:

1.Provide a better explanation for Fig. 2, including describing the protocol used to obtain these results in the Materials and Methods section.

2.The authors stated that “It is likely that at clinically effective concentrations of TIQ-15, the CXCR4 density is minimally affected” (line 319-320). The authors may want to revise their statement based on their own data: TIQ-15 concentration needs to be >10 µM to suppress HIV-1 replication to an undetectable level (Fig. 3D), and at this concentration TIQ-15 caused a significant shift in CD4 density (Fig. 4A).

**Part II – Major Issues: Key Experiments Required for Acceptance**

Reviewer #1: (No Response)

Reviewer #2: (No Response)

**Part III – Minor Issues: Editorial and Data Presentation Modifications**

Reviewer #1: (No Response)

Reviewer #2: (No Response)

PLOS authors have the option to publish the peer review history of their article (what does this mean?). If published, this will include your full peer review and any attached files.

Reviewer #1: No

Reviewer #2: No

---

## [Editor Report · Acceptance letter]

8 Aug 2024

Dear Dr. Wu,

We are delighted to inform you that your manuscript, "Characterization of a CXCR4 antagonist TIQ-15 with dual tropic HIV entry inhibition properties," has been formally accepted for publication in PLOS Pathogens.

Best regards,

Michael Malim

Editor-in-Chief

PLOS Pathogens

orcid.org/0000-0002-7699-2064